# Decadal changes in Atlantic overturning due to the excessive 1990s Labrador Sea convection

C. W. Böning [1,2] ✉, P. Wagner[1], P. Handmann [1,3], F. U. Schwarzkopf [1], K. Getzlaff[1] & A. Biastoch [1,2]

Changes in the Atlantic Meridional Overturning Circulation (AMOC) represent a crucial component of Northern Hemisphere climate variability. In modelling studies decadal overturning variability has been attributed to the intensity of deep winter convection in the Labrador Sea. This linkage is challenged by transport observations at sections across the subpolar gyre. Here we report simulations with an eddy-rich ocean model which captures the observed concentration of downwelling in the northeastern Atlantic and the negligible impact of interannual variations in Labrador Sea convection during the last decade. However, the exceptionally cold winters in the Labrador Sea during the first half of the 1990s induced a positive AMOC anomaly of more than 20%, mainly by augmenting the downwelling in the northeastern North Atlantic. The remote effect of excessive Labrador Sea buoyancy forcing is related to rapid spreading of mid-depth density anomalies into the Irminger Sea and their entrainment into the deep boundary current off Greenland.

The concept of the AMOC represents a useful condensation of the three-dimensional current systems in the Atlantic Ocean that conspire in transporting warm, buoyant waters north and colder, denser waters south[1–3]. The mechanisms of variations in the strength of the AMOC are under debate, especially on decadal time scales and beyond where they can have critical impacts on climate[4]. Decadal AMOC changes in the subtropical North Atlantic have been linked to density anomalies emerging in the subpolar gyre and travelling southwards along the deep western boundary of the ocean basin[5]. Modelling studies pointed to the emergence of meridionally coherent, low-frequency AMOC anomalies in response to low-frequency surface buoyancy forcing over the subpolar region[3,6,7]. More specifically, ocean model simulations forced by atmospheric reanalysis products, while producing diverging results with respect to multi-decadal trends, broadly concur on the AMOC variations during the last decades[8]: they typically show a strengthening during the 1980s and 1990s, followed by a decline during the succeeding decades. These decadal changes appear broadly consistent with the continuous observations in the subtropical North Atlantic since 2004[9] and various reconstructions for the last decades[3,10,11].

Previous studies have focused on the deep winter convection in the Labrador Sea[12] as a main conduit for surface buoyancy flux anomalies to the deep ocean[13], emphasising the impacts of periods of predominantly positive and negative indices of the North Atlantic Oscillation on subsurface density changes[14] and their repercussions for the AMOC[3,7]. However, the supposition of a causal relationship between Labrador Sea convection and decadal AMOC variability is disputed because of its apparent incompatibility with various observational findings: the overturning circulation inferred from hydrographic sections across the Labrador Sea appeared rather weak even during the period of intense deep convection in the 1990s[15]; no traces of changes in convection intensity could be identified in multi-year records of the deep western boundary current in the Labrador Sea[16]; and the observed slackening of the AMOC in the subtropical North Atlantic since 2008 occurred in the lower North Atlantic Deep Water, with no detectable decadal change in the transport of the Labrador Sea Water (LSW) layer[9]. Moreover, the first results from the Overturning in the Subpolar North Atlantic Programme (OSNAP) observing system for the period 2014–2018 demonstrated that the bulk of the downwelling

[1]GEOMAR Helmholtz Centre for Ocean Research Kiel, Kiel, Germany. [2]Faculty of Mathematics and Natural Sciences, Christian-Albrechts Universität zu Kiel, Kiel, Germany. [3]Lhyfe, Nantes, France. ✉e-mail: cboening@geomar.de

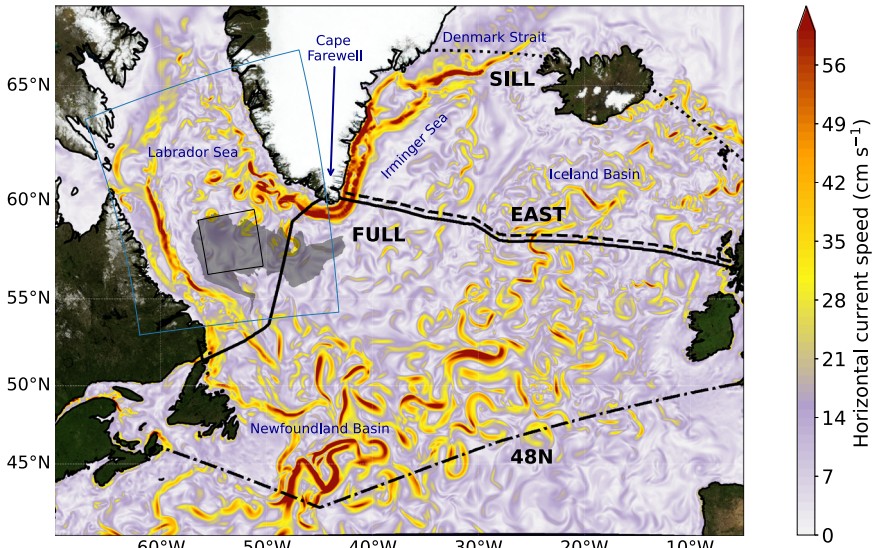

**Fig. 1 | Circulation of the subpolar North Atlantic.** Snapshot of surface speed in the high-resolution model VIKING20X, illustrating the meandering flow of the North Atlantic Current and the narrow boundary current emerging south of the Denmark Strait along the eastern continental shelf of Greenland. Shaded in grey is the area where convection exceeded 1800 m depth during the winters of 1990–1994. The black lines indicate the trans-Atlantic transport sections denoted SILL (dotted), FULL (solid), and its eastern portion EAST (dashed), as well as 48N (dashdot). The blue frame encloses the area for which the annually repeated heat flux is applied in experiment SENS. The black frame denotes the area of averaging for the water mass properties shown in Fig. 2.

limb of the AMOC occurs in the Irminger and Iceland basins, with a very weak additional contribution from the Labrador Sea[17,18].

We have investigated the patterns and variability of the overturning circulation in the subpolar North Atlantic by conducting a set of simulations with a global ocean-ice model configuration (VIKING20X; ref. 19 and see the "Methods" section), employing a nesting approach with a refined grid spacing of 0.05° over the Atlantic Ocean (corresponding to a mesh size of ~3 km at 60°N) to capture the energetic mesoscale flow patterns in the subpolar North Atlantic (Fig. 1). Of particular relevance for the formation and export of deep water is the realistic representation in this high-resolution model of the eddy activity in the Labrador Sea[20] which represents an important constraint for the extent and intensity of the deep winter convection[21,22], and of the complex structure of the deep boundary current system[16,19]. The finer representation of steep topographic slopes also supports the simulation of critical aspects of the abyssal circulation such as the outflows from the Nordic Sea[19,23] that provide the densest source waters for the lower limb of the AMOC.

Our aim is to re-assess the role of the Labrador Sea buoyancy fluxes in interannual to interdecadal AMOC changes and their geographical distribution in the subpolar North Atlantic. A particular focus is on the effect of the period of strongly enhanced winter heat losses related to the extended positive phase of the North Atlantic Oscillation (NAO) during the late 1980s to mid-1990s which was responsible for the emergence of an exceptionally dense vintage of LSW[14,24,25]. In order to isolate the effect of this outstanding episode in the Labrador Sea forcing and LSW generation on the AMOC we compare the transport variability of two experiments: a hindcasting experiment (CTRL) using the full set of ocean forcing data based on the JRA55-do products[26] through 1958–2019; and a sensitivity experiment (SENS) in which the year-to-year variations in the buoyancy forcing of the Labrador Sea area were artificially constrained. Experiment SENS branches off from CTRL in May 1980, and was run through 2019 under the same forcing as CTRL, except for the heat fluxes over the Labrador Sea (between 53°N and 65°N, and from 45°W to 64°W; as depicted in Fig. 1) which were replaced by a repetitive application of the fluxes from the period May 1980 to April 1981 (see the "Methods" section; Supplementary Fig. 1). Some assessment of the robustness of the LSW formation and overturning variability simulated in CTRL is provided by a

supplementary set of three hindcasting experiments with VIKING20X, with different choices for the atmospheric forcing, freshwater fluxes and continental runoff (including the meltwater fluxes from Greenland), and initial conditions (Supplementary Table 1).

## Results

### Decadal changes in LSW formation

The prime consequence of the forcing perturbation is the suppression of interannual variations in the intensity of deep winter convection in the Labrador Sea (Fig. 2a). As intended, SENS particularly succeeds in removing the prominent signature in CTRL of the exceptionally strong and deep convection events during the first half of the 1990s, while retaining a weaker level of convection intensity corresponding to the negative NAO phases before and after that period.

Of particular relevance to the dynamics of the subpolar circulation is the history of changes in the density of the convectively formed water. While over much of the hydrographic record of Labrador Sea Water (LSW), the effect of changes in temperature on the density tended to be compensated by changes in salinity[14,27], the intense convection winters of 1990–1994 stood out as a formation period of an exceptionally dense vintage of LSW[14,24,25]. This event is reproduced in CTRL, with the emergence and strong volumetric increase of an anomalously dense variety of LSW during the intense convection period of the 1990s (Fig. 2b and Supplementary Fig. 2; cf. Fig. 6a, and ref. 22 for comparison with observational records). The density and volume of the LSW produced during this period were not reached during any time after 1995 in CTRL; in particular, there was only a comparatively small increase in the density of the waters formed during the enhanced convection phase after 2014. The production of the dense class of LSW is effectively curbed by the perturbation in the local heat fluxes in SENS (Fig. 2c), with relatively little changes in the volume of the upper LSW during the course of the experiment. The pair of simulations with and without these decadal changes of deep winter convection thus provides a useful means to address the lingering question of its causal relationship with changes in the subpolar overturning circulation.

### Decadal overturning changes in the subpolar basins

The mean (1980–2019) strength of the AMOC in CTRL exceeds 18 Sv (Sverdrups; 1 Sv = 10⁶ m³ s⁻¹) in the latitude range 47°–58°N, with a

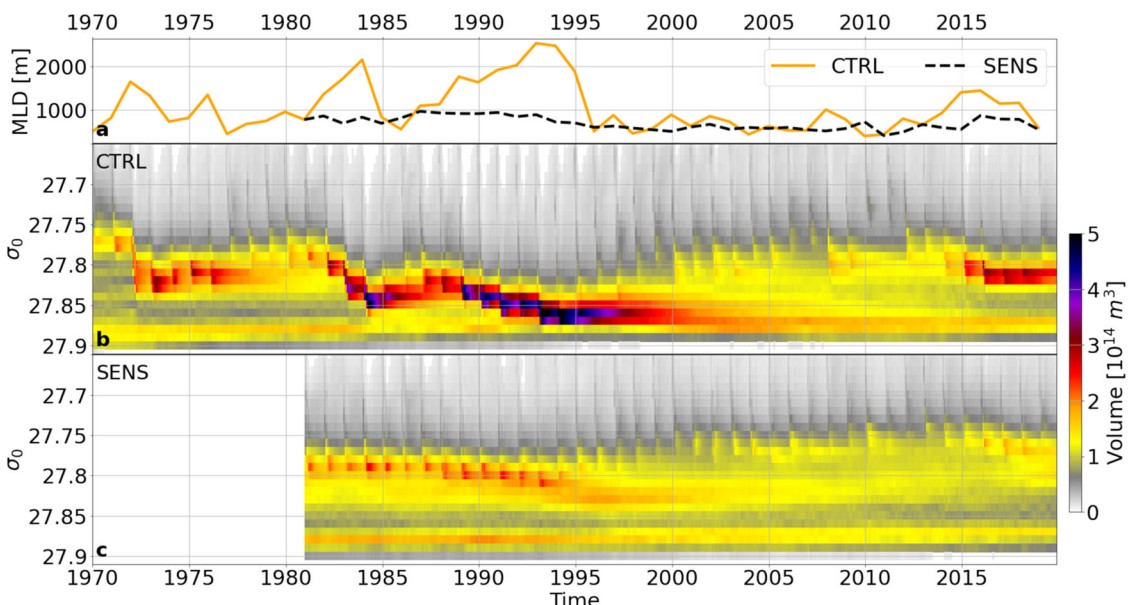

**Fig. 2 | Variation of water mass formation in the Labrador Sea. a** Depth of the winter mixed layer (MLD) averaged over the Labrador Sea between 56.5°N–59.3°N and 56.0°W–50.8°W as shown in Fig. 1, for the hindcasting simulation CTRL (orange) and the sensitivity experiment with the repeated surface heat fluxes of 1980/81 (SENS; black). **b** Volume of water in potential density bins of 0.01 kg m⁻³ for CTRL, showing the emergence of the exceptionally dense vintage of Labrador Sea Water during the intense convection period of the 1990s. **c** The evolution of the same quantity in SENS where the formation of this water mass is suppressed.

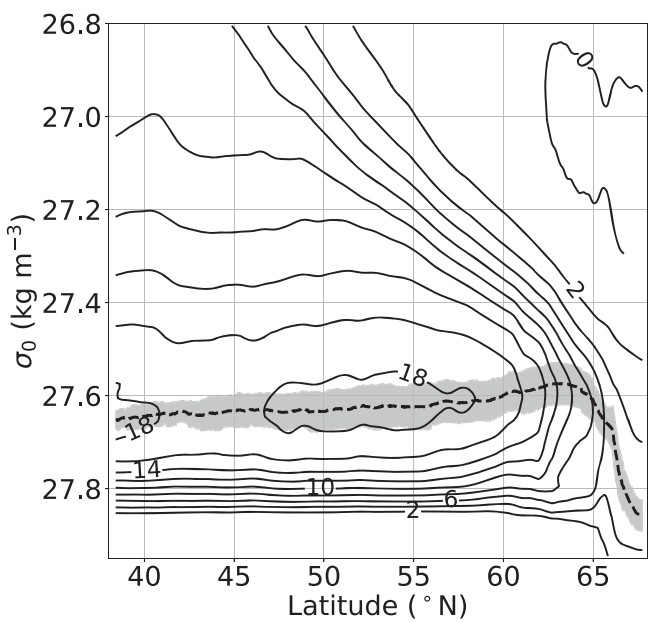

**Fig. 3 | Meridional overturning circulation in the subpolar North Atlantic.** Streamfunction of the zonally integrated meridional volume transport (AMOC) as a function of latitude and potential density for experiment CTRL, averaged over the period 1980–2019. The dashed line depicts the density dividing the northward ('upper') and southward ('lower') limbs of the overturning cell, the shading indicates the standard deviation of its interannual variability; the contour interval is 2 Sv (1 Sv = $10^6$ m³ s⁻¹). See Supplementary Fig. 4 for more detailed transport profiles.

maximum of 18.5 Sv at 54°N (Fig. 3). The zonally integrated view illustrates the gradual increase in density of the northward flowing subtropical waters in the upper limb of the overturning cell until all flow turns southward for potential densities exceeding ~27.6 kg m⁻³. Most of the mass flux across that density threshold takes place north of 60°N, with a comparatively minor gain of the lower limb transport of

about 2 Sv at latitudes between 60°N and 52°N. There are only small diapycnal mass fluxes at mid-latitudes, so the mean AMOC at sub-tropical latitudes (e.g., 18.1 Sv at 26°N) remains close to the transport established in the subpolar North Atlantic (Supplementary Fig. 3).

A further geographical deconstruction of the 'downwelling' limb of the AMOC (the diapycnal mass flux from the northward to the southward limb of the AMOC) and its temporal evolution in both simulations is obtained by building on the observational strategy of the OSNAP programme[17]. Figure 4a depicts the overturning transports for the set of sections indicated in Fig. 1: the full transatlantic section of OSNAP ('FULL') and its eastern portion ('EAST') are augmented here by a section along the Greenland–Iceland–Scotland-Ridge ('SILL') to obtain the contribution of the overflows and by a section at the transition from the subpolar to the subtropical gyre ('48N'; chosen to follow the AR19/A2-section of the World Ocean Circulation Experiment). In accordance with the zonally integrated picture of Fig. 3, there is only a little difference between the transports at FULL and 48N, demonstrating that almost all downwelling is accomplished north of FULL.

Of major interest then are the relative contributions of the northeastern basins and the Labrador Sea. While the former is given directly by EAST, the latter cannot be gleaned in a simple way from the overturning across the western portion of FULL ('WEST'), without accounting for its more complex distribution and partial compensation of the transports in density space (ref. 17; Supplementary Fig. 4). More specifically, the overturning cell in the Labrador Sea adds to the lower limb transport at EAST only for densities above 27.68 kg m⁻³, while reducing it for lighter densities. Its net contribution to the transatlantic overturning cell (FULL) is therefore mainly a shift in the density profile of the lower limb, i.e., an increase in the density of the streamfunction maximum from 27.53 kg m⁻³ at EAST to 27.64 kg m⁻³ at FULL (in accord with the observational account[17]). Instead of using WEST, the net contribution to the AMOC by downwelling in the Labrador Sea can therefore more directly be inferred from the difference between FULL and EAST. For the period of the first 4-year phase of OSNAP (2014–2018), the transports simulated in CTRL are somewhat weaker than the observational products[18] for both FULL (15.3 vs. 16.6 Sv) and EAST (12.8 vs. 16.8 Sv). The gain between EAST and FULL is not completely negligible as in the observational account

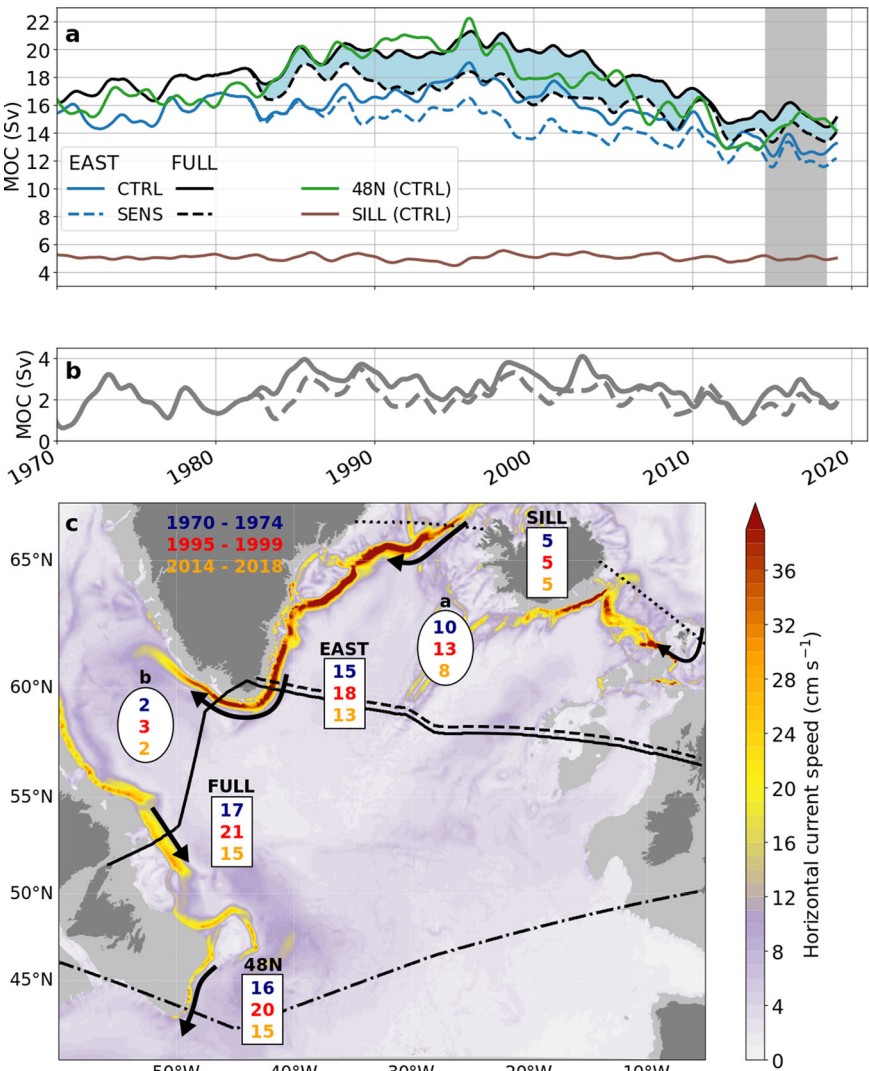

**Fig. 4 | Geographical deconstruction and inter-decadal variation of the overturning circulation. a** The overturning transports at sections FULL and EAST (for experiments CTRL and SENS), SILL and 48N (CTRL only for clarity: SENS does not differ from CTRL at SILL, and the difference between SENS and CTRL at 48N is similar to that difference at FULL); the blue shading highlights the difference between CTRL and SENS at FULL; the grey shading indicates the first phase of the OSNAP observations. **b** FULL minus EAST transport for CTRL and SENS (dashed), indicating the net contribution to the basin-wide overturning cell (AMOC) by downwelling in the Labrador Sea. **c** Map summarising the evolution of the lower limb of the AMOC for three different 5-year periods of CTRL: section transports (values in rectangles; in Sv) and diapycnal downwelling from the upper to the lower limb (values in ovals; in Sv) implied by the transport divergence between EAST and SILL (oval 'a'), and FULL and EAST (oval 'b'). The horizontal current speed averaged between the potential density surface 27.55 kg m$^{-3}$ and the bottom (colour bar; in cm s$^{-1}$) illustrates the concentration of the lower limb transport along the continental slopes of Greenland and North America. The transport evolves from two main sources, the outflow from the Nordic Seas and its enhancement by downwelling in the northeastern basins between SILL and EAST. After minor augmentation during its passage through the Labrador Sea, it reaches its maximum strength near 52°N (cf. Fig. 3). The increase of ~4 Sv at FULL and 48N between the 1970s and 1990s originated mainly from enhanced downwelling between SILL and EAST, with a contribution of ~1 Sv from enhanced downwelling in the Labrador Sea.

(where there is no significant difference between FULL and EAST); however, the difference of only about 2 Sv during this period still concurs with the major conclusion of the observations[17,18] that the net overturning transport across the full transatlantic section is governed by the diapycnal downwelling in the northeastern subpolar North Atlantic, north of the EAST section.

A prominent feature of the transport time series in CTRL (Fig. 4a), holding for all EAST, FULL, and 48N, is the increase from the 1970s to a maximum overturning during 1994–1998, followed by a decline thereafter (see Supplementary Fig. 5 for a comparison of CTRL with the additional set of experiments). The amplitude of these inter-decadal changes is similar at 48N and FULL, with an increase by ~4 Sv and a decline by ~5 Sv. Remarkably, the changes in the overturning at EAST are not much smaller (about 3 and 4 Sv, respectively) than those at the full transatlantic sections. Since the SILL transport has remained

steady, most of the inter-decadal variation can thus be attributed to the downwelling in the Irminger and Iceland Seas.

A more detailed view of the contribution to the changes in the total (FULL) overturning by downwelling in the Labrador Sea is provided in Fig. 4b. For CTRL a similar mean appendage of around 2 Sv as noted for 2014–2018 appears characteristic for the periods before the mid-1980s and after 2005. During the intermediary period of enhanced overturning the Labrador Sea contribution was enhanced by about 1 Sv. A small enhancement of similar magnitude is exhibited in the difference between CTRL and SENS during this period, confirming that during the strong cooling period, there were only weak repercussions for the AMOC by an increased local downwelling in the Labrador Sea. Another interesting aspect of the time series in Fig. 4b is that the amplitude of the interannual variability is similar in CTRL and SENS, suggesting that the Labrador Sea contribution to the

AMOC is not sensitive to the local buoyancy forcing on these time scales.

The geographical distribution of the section transports and their inter-decadal variation in CTRL is summarised in Fig. 4c, by illustrating the evolution of the lower limb of the AMOC from the Nordic Seas outflow in the north (SILL) to its export from the subpolar gyre (48N). Along the continental slopes of Greenland and Labrador, the bulk of the southward transport is concentrated in the deep western boundary current. The succession of section transports may hence be viewed in terms of a progressive increase in the strength of the AMOC-related portion of the deep boundary current system, i.e., the portion not being part of recirculation cells within the subpolar gyre: induced in the northern Irminger Basin by the combined outflows from the Nordic Seas (SILL), the transport is strongly increased during its progression along the continental slope of Greenland until the deep water exits the Irminger Sea at Cape Farewell. The downwelling between SILL and EAST also governs the inter-decadal changes in the deep export. After only modest enhancement in the Labrador Sea during both weak and strong convection periods, the southward transport has reached its near-final strength when passing the western end of FULL; subsequent changes farther downstream between FULL and 48N appear negligible (cf. Figure 3).

## Remote effect of the excessive 1990s' Labrador Sea cooling period

The absence of a significant contribution to the inter-decadal AMOC changes in the Labrador Sea does, however, not imply a minor impact of the regional buoyancy forcing changes. An assessment of their impact is facilitated by inspecting the differences between CTRL and SENS in the evolution of the subpolar overturning circulation. Figure 4a shows that in SENS nearly all of the increase characterising the section transports for the 1980-to-mid-1990s period in CTRL is eliminated, demonstrating that the surface heat flux in the Labrador Sea had a leading impact in generating the AMOC peak in the 1990s. Its influence is less obvious for the subsequent, declining trend of the AMOC in CTRL which is reduced only weakly in SENS. The smaller difference between the trends of CTRL and SENS points to the influence of other factors, common to CTRL and SENS, that mask the influence of the Labrador Sea heat flux during this phase (see Supplementary Fig. 7).

The major impact of the Labrador Sea heat flux on the inter-decadal evolution of the AMOC exposed here stands in contrast to its negligible influence on the interannual AMOC variability: there is very little difference in the interannual transport variability between CTRL and SENS, with correlations (band-pass filtered between 1 and 5 years)

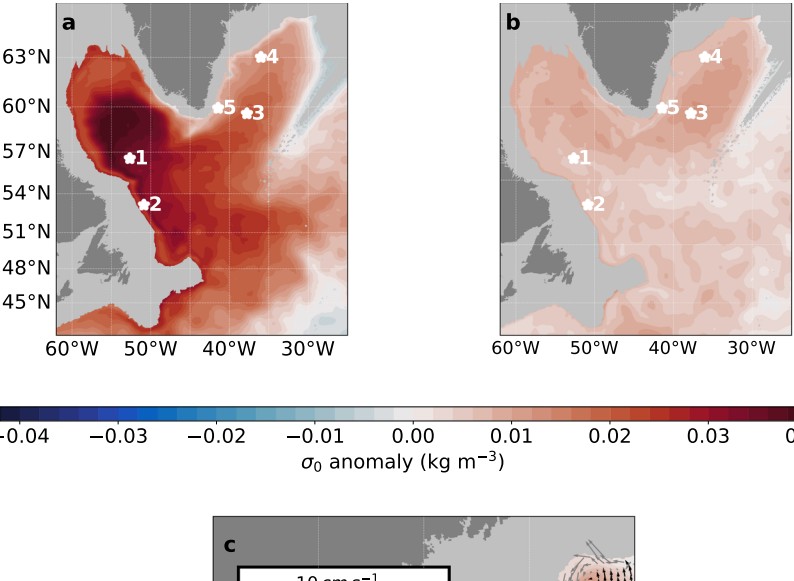

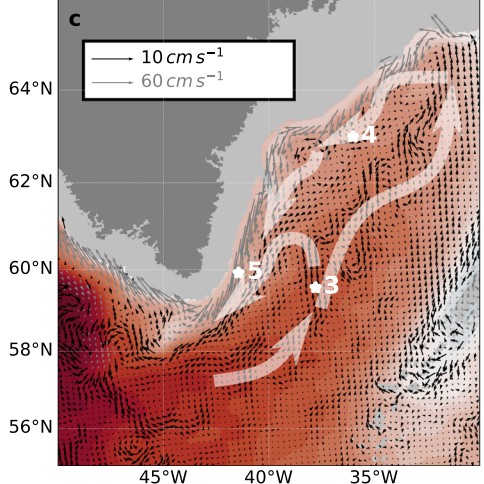

**Fig. 5 | Spreading of the Labrador Sea density anomaly.** Potential density anomaly **a** in experiment CTRL and **b** in experiment SENS averaged over the period 1990–1995 at 1500 m depth ($\sigma_0$; kg m$^{-3}$; colour shading) with the positions of stations (white stars) for which time series are shown in Fig. 6. The positions 1, 2, 3, and 5 correspond to moorings of the observational programme OSNAP (K1, K9, M5 and M1, respectively); position 4 represents a site near the origin of the deep western boundary current in the northern Irminger Basin, about 250 km south of Denmark Strait. **c** Depiction of the current field responsible for the advective spreading of water from the Labrador Sea into the Irminger Sea in CTRL, with a schematic accentuation of the main pathway.

of $r = 0.95$ (0.87) for EAST (FULL). Since the SILL transport is not altered between CTRL and SENS, this points to a leading role of the buoyancy forcing in the Irminger and Iceland basins in the generation of this variability, consistent with observational accounts of the local air–sea fluxes and water mass transformation over this area[28].

The remote effect of the Labrador Sea forcing on the generation of the AMOC peak in the mid-1990s is facilitated by the rapid spreading of LSW into the Irminger Sea by the mid-depth circulation. Figure 5 shows that the large volume of exceptionally dense waters built-up in CTRL (cf. Supplementary Fig. 6 for the other hindcast simulations) during the winters of 1990–1994 got dispersed from the convective region along three main pathways known from classical hydrographic studies[29–32]: in addition to a narrow southward tongue along the deep western boundary current (the most rapid export route for near-boundary convection signals[33]), and an eastward branch at 50°–53°N with the North Atlantic Current, there is also a conspicuous north-eastward extension of the LSW anomaly into the Irminger Sea. The existence of this direct and fast path from the Labrador Sea to the Irminger Basin had been substantiated by float observations: first at 700 m depth[34], and more recently for the Labrador Sea water density range at 1000–1500 m[35], the float trajectories revealed a series of elongated, peripheral recirculation cells straddling the offshore shoulder of the boundary current system.

The spreading pattern is manifested in the temporal evolution of the LSW layer thickness in a sequence of positions through the Labrador and Irminger Sea (Fig. 6). A prominent signature of the anomaly building up in the vicinity of the convective patch (position 1) immediately emerges in the deep western boundary current of the Labrador Sea (position 2, indicative of the fast export pathway described by ref. 33. However, the anomaly is rapidly conveyed also, with a lag of no more than a year, eastward into the southern Irminger Basin (position 3) via the interior currents illustrated in Fig. 5c. Similar to the float observations[35], the model indicates an extension of the mid-basin current into the northern Irminger Sea. Together with the deep northward boundary current along the western flank of the Reykjanes Ridge which is fed, in part, from the

outflows across the Iceland–Scotland Ridge, the mid-basin flow forms a cyclonic circulation in the Irminger Basin fuelling the southward western boundary current off Greenland. While there is a gradual erosion and retardation of the original LSW peak indicative of a progressive mixing with ambient waters, an imprint of the anomaly can still be identified in the western boundary current just south of the Denmark Strait outflow regime (position 4). Due to the mixing with ambient waters, the anomaly becomes further atte-nuated along the course of the boundary current, but can still be traced into its southern reaches near Cape Farewell (position 5).

## Density changes in the deep boundary current along south-eastern Greenland

An intriguing aspect of the inter-decadal density evolution in the deep boundary current off Greenland is its correspondence to the over-turning changes at EAST (Fig. 7), suggesting that the downwelling into the lower limb has been modulated by the thick layer of extremely dense LSW invading the Irminger Basin and its western boundary regime during the mid-1990s. The model behaviour appears in accord with recent analyses of climatological hydrographic data sets which pointed to a concentration of diapycnal downwelling connecting the upper and lower limbs of the AMOC along the continental slope of Greenland[36]. The abyssal density in the boundary current is deter-mined by the properties of the Nordic Seas outflow and its mixing with the ambient intermediate waters, resulting in a drop in the density and an increase in the transport primarily in the downslope flow regime south of the sill[37]. However, intensified diapycnal mixing rates have also been found in the boundary current near Cape Farewell[38], sug-gesting a continuous entrainment of intermediate waters into the lower limb along the western boundary of the Irminger Sea. A perti-nent finding of measurement programmes in the Denmark Strait is that both the transport and density of the overflow have remained rather constant during the last decades[39]. This also holds for the pre-sent simulations, except for a decreasing trend in the sill density emerging during the last 1–2 decades of both CTRL and SENS (Sup-plementary Fig. 7). The increase of the deep boundary current density

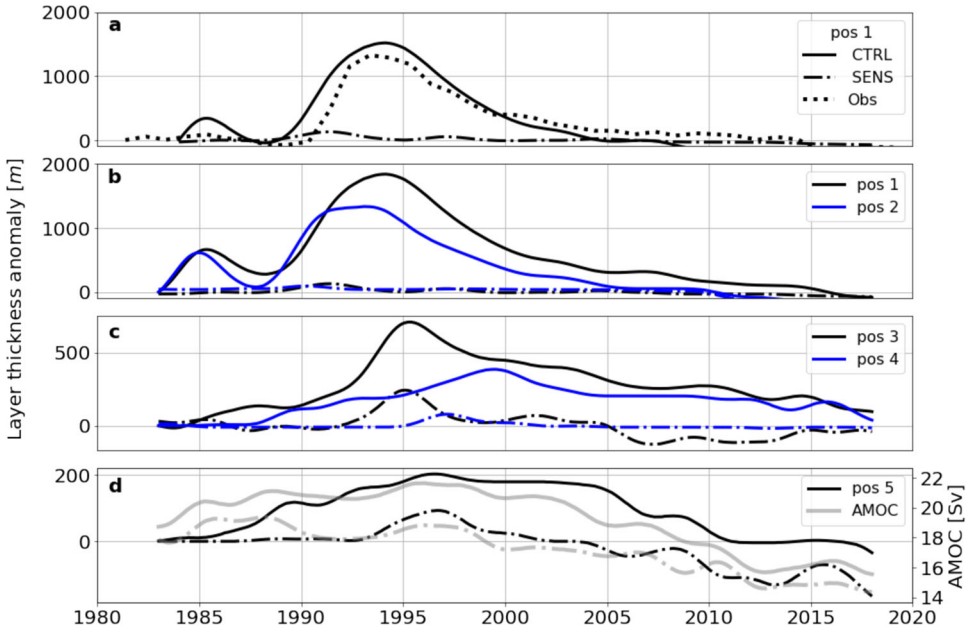

**Fig. 6 | Evolution of Labrador Sea Water (LSW) thickness anomalies. a–d** Layer thickness anomaly (relative to January 1983 in the filtered time series) of the dense class of LSW renewed in experiment CTRL during the intense 1990s convection period, for the five stations denoted in Fig. 5. The evolution in CTRL (continuous lines) is contrasted with experiment SENS (dash-dotted) which lacks the thickness peak in the 1990s. **a** The anomaly just south of the main convection area (pos. 1) is shown in comparison to the observational product of ref. 25 (dotted line). **d** The evolution of the LSW anomaly in the boundary current (pos. 5; black lines) is shown along with the overturning transport at FULL (grey lines) for both CTRL and SENS.

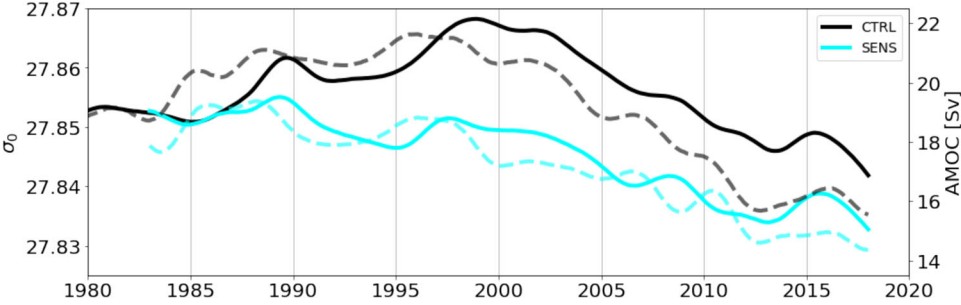

**Fig. 7 | Overturning transport in relation to wbc density changes.** Inter-decadal evolution of potential density in the deep western boundary current at position 5, averaged over 1500–2000 m depth (continuous lines), and overturning transport (AMOC) at section FULL (dashed lines) for experiments CTRL and SENS.

near Cape Farewell to maximum values in the mid-1990s, and its co-evolution with the overturning at EAST, thus suggest that the downwelling into the lower limb has been modulated in the model by the extremely dense LSW invading the Irminger Basin and its western boundary regime during the mid-1990s.

## Discussion

Previous discussions of potential dynamical impacts of (multi-)decadal variations in the Labrador Sea convection intensity mainly considered the southward export of anomalies and their contribution to mid-depth density gradients in the western Newfoundland Basin (e.g., refs. 2,3,5). Our study provides a different perspective by highlighting the role of the fast propagation of density anomalies into the Irminger Sea. A similar pathway was identified for upper layer (0–1000 m) density anomalies, enabling a contribution of Labrador Sea signals to interannual Irminger Sea density variations[40]. The present simulations suggest, however, that an effect of LSW variability on the overturning was confined to the extended period of excessive winter cooling associated with the prominent, positive phase of the NAO from the late 1980s to the mid-1990s. A unique characteristic of this period was the production of a large volume of exceptionally dense LSW, whereas other LSW anomalies associated with the interannual convection variability during the last six decades remained too small to affect the downwelling. The present ocean model hindcast concurs in this respect with a 300-year pre-industrial climate simulation which suggested a dominant influence on the AMOC by large Labrador Sea buoyancy forcing anomalies on multi-decadal time scales[41].

An intriguing aspect of the forcing process exposed here is that it involves a modulation of the density of the abyssal waters along the East-Greenland continental slope which constitute the source of the lower North Atlantic Deep Water. This raises the possibility that LSW density anomalies may manifest themselves in the transport of deeper portions of the lower AMOC limb, rather than in the upper portion encompassing the LSW density range. The mechanism thus offers a possible explanation for the puzzling aspect of the AMOC observations at 26.5°N (ref. 9) which recorded a declining trend in the lower deep-water range after 2008, apparently at odds with the lack of changes in its primary source, the overflow from the Nordic Seas (ref. 39). It also has potential ramifications for the latitudinal coherence of (multi-) decadal AMOC anomalies, since a propagation from the subpolar to the subtropical gyre has been described only for the lower, but not the upper layers of the deep water[42].

The emerging picture is in accord with classical accounts of the deep circulation in the subpolar North Atlantic. The composition of pioneering velocity measurements between the Denmark Strait and Cape Farewell suggested that the transport of the dense water masses spilling across the sills is more than doubled by the entrainment of ambient water along the continental slope of Greenland[43]. The crucial role of the outflow of dense water and its entrainment of intermediate water in sustaining the lower limb of the AMOC was confirmed in a recent study of heat and freshwater budgets[44]. Our model simulation adds to these views by drawing attention to the impact of decadal density anomalies in the mixing product of the overflow with the ambient waters in the Irminger Basin. A major influence of the outflow density on the AMOC had been noted in regional model studies in which variations in the density of the outflow product were prescribed by lateral boundary conditions[45–47]. However, the outflow density appeared as a leading factor only if it clearly exceeded the LSW density[48]. Meeting this condition presents a formidable challenge to the global ocean and climate models in which that density difference tends to be diminished by an overproduction of LSW[49] on the one hand and a spurious dilution of the overflow water on the other hand[23,50]. Recent studies suggested the need for very fine grid sizes for realistic representations of the small-scale processes governing the deep formation in the subpolar North Atlantic[51]; however, some sensitivities to numerical choices and para-meterisations remain even at 1/20°-resolution[19,52]. A systematic investigation of the factors influencing the evolution of the AMOC on inter-decadal and longer time scales thus represents a challenging task, calling for a coordinated, multi-model programme of experimentation that would need to include models of very fine resolution.

## Methods
### Model experiments
VIKING20X is a global ocean-sea ice model configuration based on the Nucleus for European Modelling of the Ocean (NEMO) system, code version 3.6, incorporating the LIM2 sea ice model[53]. It builds on the widely used "eddy-present" (0.25° horizontal grid) global ocean-sea ice configuration ORCA025 by way of a two-way nesting scheme, Adaptive Grid Refinement in FORTRAN (AGRIF)[54], using a grid refinement by a factor of 5 between 34°S in the South Atlantic and ~70°N in the northeastern North Atlantic. While the horizontal mesh size of ORCA025, about 15 km in the Labrador Sea, is not sufficient to capture mesoscale eddy processes governed by the Rossby radii of about 10 km in that region, the grid refinement provides an "eddy-rich" nest domain with mesh sizes of 2–3 km around Greenland. The model uses 46 geopotential z-levels in the vertical, with layer thick-nesses from 6 m at the surface gradually increasing to ~250 m in the deepest layers. The bottom topography of the nest is based on ETOPO1[55] and represented by partially filled cells with a minimum layer thickness of 25 m allowing for an improved representation of near-bottom flows[56]. A discussion of further technical details, including numerical advection and diffusion schemes as well as the bottom and sidewall boundary conditions is given in ref. 19; it also provides a comprehensive account of studies documenting the representation of subpolar North Atlantic circulation patterns with this model configuration. For the reproducibility of the experiments presented here, the model code and complete namelists are published alongside this study in the institutional repository ref. 57.

For atmospheric forcing, river runoff and freshwater fluxes from ice sheets, we used JRA55-do v1.4 for the period 1958–2019, developed

by ref. 26 as a common forcing product for ocean hindcasts and model intercomparison studies. The integration in CTRL (referred to as VIKING20X-JRA-OMIP in ref. 19) followed the OMIP-2 protocol[58], starting from rest and initial temperatures and salinities of the World Ocean Atlas 2013 (WOA13)[59]. To reduce spurious model drift that may potentially result even from small errors in the forcing or missing feedback from the atmosphere, the integrations adhered to the common practice in global ocean modelling by employing weak damping of sea surface salinity to the initial climatological field, with a relaxation time scale of 1 year for a 50 m surface layer (piston velocity 50 m/yr). Damping was not applied in sea-ice-covered regions, and also not in an 80 km wide band around Greenland to allow for a free evolution of the very fresh waters on the continental shelf.

The sensitivity experiment SENS branches off from CTRL in May 1980 and was run through 2019 under the same forcing as CTRL, except for the heat fluxes over the Labrador Sea (between 53°N and 65°N, and from 45°W to 64°W) which were replaced by a repetitive application of the fluxes of the period from May 1980 to April 1981. The choice of the period 1 May 1980–30 April 1981 is based on two considerations. First, on the intention to selectively suppress the effect of the strong increase in convection activity during the 1980s and early 1990s with minimal distortion of other forcing elements impacting the evolution of the AMOC; second, on minimising disturbances involved in step function changes in the day-to-day variation in forcing. The first criterion appears optimally met by the years 1980–1981 representing the transition of the North Atlantic Oscillation index from a prolonged negative phase to the increasingly positive phase of the 1980s and 1990s. This is reflected by the evolution of key forcing elements in CTRL, specifically the heat fluxes and convection intensity, which for the winter of 1980–1981 were close to their multi-decadal means (Supplementary Fig. 1). For the second we followed the development of a 'Repeated Year Forcing' scheme for JRA55-do by ref. 60 who found that a step from 30 April to 1 May provided the smoothest transition period possible. As seen in Figs. 2 and 5, the choice of the repeated 1980/81 forcing period succeeded in suppressing the formation of the extremely dense LSW during the 1980s–1990s; an inspection of its behaviour in comparison to CTRL thus provides a useful means to isolate the impact of this Labrador Sea forcing event on the generation of the AMOC peak in the 1990s. It should be acknowledged, however, that further interpretation of the AMOC evolution in SENS is limited by possible quantitative sensitivities to the artificial experimental design: e.g., it is conceivable that a choice of the repeated year with different intensity of winter convection could have an effect on the interdecadal trend of the perturbation experiment.

A supplementary set of three hindcast experiments is employed to allow some assessment of the robustness of the decadal variability simulated in CTRL. These experiments (denoted CORE, JRA-cr and JRA-short) differ in choices for the initial conditions, the atmospheric forcing and the freshwater runoff and surface fluxes (Supplementary Table 1). Experiment CORE is initialised in 1958 after a 30-year spin-up and forced by the CORE v2 data-set[61,62]. Freshwater runoff is provided by a monthly climatology, it therefore lacks the increasing meltwater trend from Greenland; precipitation is prescribed by an inter-annually varying monthly field. Experiment JRA-cr branches off from CORE in January 1980, using similar forcing data (JRA55-do v1.3) as CTRL, except for the runoff, which builds on the same climatological product as used for CORE. Experiment JRA-short is subject to the same forcing as CTRL, but initialised in 1980, in the same manner as JRA-cr. In all of these experiments, sea surface salinity relaxation (SSSR) is weaker compared to CTRL with a timescale of 4.11 years for a 50 m surface layer (piston velocity 12.2 m/yr); and in CORE and JRA-cr no mask with suppression of SSSR around Greenland is applied. For further details of parameterisation choices in the VIKING20X experiments, we refer to the institutional repository ref. 57.

## Meridional overturning circulation

The transport of the Atlantic MOC across individual sections is defined as the maximum of the streamfunction in density space

$$\mathrm{MOC}(t) = \max(\Psi(\sigma, t)) = \max\left(\int_{\sigma_{max}}^{\sigma}\int_{e}^{w} \mathbf{v}(x, \sigma, t)\mathrm{d}x\,\mathrm{d}\sigma\right) \quad (1)$$

where $\mathbf{v}$ is the cross-section velocity, and the integration extends from the eastern ($e$) to the western ends ($w$) of the sections and from the densest layer ($\sigma_{max}$) up to a certain density ($\sigma$) with d$\sigma$ referring to the layer thickness between $\sigma$ and $\sigma_{max}$. Note that the density at which the MOC reaches its maximum varies over time. Apart from the direction of integration, this is consistent with the procedure described for the OSNAP array[17,18].

## Mixed layer depth

The depth of the mixed layer (MLD) is defined here as the depth at which the potential density referenced to the surface ($\sigma_0$) has changed by 0.01 kg m$^{-3}$ relative to 10 m depth (see ref. 21 for a comparative discussion with alternate approaches). The temporal evolution displayed in Fig. 2 represents a time series of the maximum MLD in winter averaged over the Labrador Sea between 56.5°N–59.3°N and 56.0°E–50.8°W.

## Analysis of volume per potential density in the Labrador Sea

The volumes in density space shown in Fig. 2 are computed with the following definitions. The Labrador Sea was defined as the area between the southern tip of Greenland (43.5°W, 60.5°N), Newfoundland (St. Johns; 53°W, 47°N), the Hudson Strait ((65°W, 58°N), (66°W, 64°N)) and the Davis Strait ((64°W, 65.5°N), (53°W, 66.5°N)). The total volume of all grid boxes in this area adds up to a total volume of 2.268688*10$^{15}$ m$^3$. The grid box volumes in the Labrador Sea were binned using two different bin sizes according to the density volume distribution: a bin size of 0.2 kg m$^{-3}$ was used for potential densities of 26.4 kg m$^{-3}$ and smaller, and for potential densities of 28.1 kg m$^{-3}$ and larger; the bin size was refined to 0.1 kg m$^{-3}$ for the density ranges 26.4–26.5 kg m$^{-3}$ and 28–28.1 kg m$^{-3}$ and further to 0.01 kg m$^{-3}$ for the central density range 26.50–28 kg m$^{-3}$.

## Analysis of potential density anomalies

The layer thickness anomalies displayed in Figs. 5 and 6 represent the vertical extent of the dense class of Labrador Sea water ($\sigma_0$ of 27.83–28.0 kg m$^{-3}$; cf. Fig. 2b) which was renewed in the model simulation (CTRL) only during the intense convection period of the 1990s. Comparison with observational reconstructions (ref. 26) suggests an offset of -0.06 kg m$^{-3}$ in the simulated Labrador Sea density fields. In the model-data comparison of the layer thickness evolution (Fig. 6a), this is accounted for by displaying the simulated layer thickness in conjunction with the thickness of the density layer 27.77–27.84 kg m$^{-3}$ from the observational product of ref. 26.

The density time series displayed in Figs. 6 and 7 and in Fig. S7b and c are computed by averaging the densities at position 5 (Fig. 5) between 1500 and 2000 m. The time series are low pass filtered with a 4-year running mean hamming filter.

## Data availability

The data and material that support the findings of this study are available through GEOMAR at https://hdl.handle.net/20.500.12085/44e539eb-5496-4298-bda5-10a4d655d34f (ref. 57). Data from the RAPID AMOC monitoring project, funded by the Natural Environment Research Council, are freely available from www.rapid.ac.uk/rapidmoc and have been accessed as version 2020.2 (ref. 63) from https://doi.org/10.5285/e91b10af-6f0a-7fa7-e053-6c86abc05a09. The observational data product to derive Labrador Sea Water thickness anomalies was updated based on ref. 25; we thank Igor Yashayaev (Fisheries and Oceans Canada, Bedford Institute of Oceanography, Dartmouth, Nova

Scotia, Canada) for providing the hydrographic data from the central Labrador Sea.

## Code availability

The NEMO code used for the numerical simulations is available at https://forge.ipsl.jussieu.fr/nemo/svn/NEMO/releases/release-3.6. Our experiments are based on revision 6721. Code modifications and namelists, as well as further information necessary to reproduce the present model simulations, are available through GEOMAR at https://hdl.handle.net/20.500.12085/44e539eb-5496-4298-bda5-10a4d655d34f (ref. 57).

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

## Acknowledgements
This research received funding from the German Federal Ministry of Education and Research (RACE Regional Atlantic Circulation and Global Change, grant no. 03F0729C). The model integrations were enabled by the provision of extensive computing resources on the HPC-systems JUWELS at the Jülich Supercomputing Centre (JSC) in the framework of the Earth System Modelling Project (ESM) and at the North German Supercomputing Alliance (HLRN). We wish to thank the DRAKKAR group for their continuous support in the model development.

## Author contributions
C.W.B. conceived the study and along with A.B. designed the model experiments. K.G., F.U.S and P.W. developed and performed the model experiments. P.H., F.U.S and P.W. produced the analyses with input from C.W.B. and A.B.; P.H and P.W. produced the figures, with contributions by F.U.S. C.W.B. wrote the manuscript; A.B., P.H., F.U.S. and P.W. contributed to the writing and refinement of the paper.

## Funding

## Competing interests
The authors declare no competing interests.
