## [Peer Review File · Nature Communications]

Decadal changes in Atlantic overturning due to the excessive 1990s Labrador Sea convectionREVIEWER COMMENTS

Reviewer #1 (Remarks to the Author):

Review Comments for "Decadal changes in Atlantic overturning due to the excessive 1990s Labrador Sea convection" by Böning et al.

This study explores the role of the Labrador Sea in affecting the inter-decadal variability of the AMOC with a high-resolution nested model. They find that by masking out the strong changes in heat fluxes of 1990s over the Lab Sea than therefore the Labrador Sea Water (LSW) production, the AMOC inter-decadal variability is affected upstream over the Irminger Sea instead of downstream over the Newfoundland Basin. The explanation they provide is that the anomalous dense Labrador Sea Water (LSW) during the 1990s invade northeastward into the Irminger Sea and get entrained into the AMOC lower limb. I found this study overall very interesting in three aspects. First, it works on a hot topic; no consensus on whether and how the Lab Sea plays a role in the AMOC variability has emerged since the OSNAP observation. Second, the results clearly show that "what happens in the Lab Sea stays in the Lab Sea" does not occur in their model; instead, the signal transits to the Irminger Sea and affects the EAST side of the OSNAP section. Finally, their explanation is reasonable and seems to be supported by observational evidence (where I have some comments as listed below). I would recommend publication of the research. I have one concern that requires some clarification and a few additional minor points.

The authors claim that the advective signal of the LSW at 1500 m is able to travel northeastward into the Irminger Sea and later get entrained into the AMOC lower limb. Meanwhile, they find the WEST has no significant difference while the EAST has a ~ 2 Sv difference between the CTRL and SENS (Fig. 4). One thing that is not clear to me, is why the export of the LSW during the peak period is not detected along WEST, but its invasion to the Irminger Sea and entrainment to the lower limb could be easily detected at EAST. In addition, if the signal is advective as shown in the model, should its northward transport across EAST and later transport back cancel out, as both directions are within the lower limb? Am I missing something? Some clarification is expected here.

Minor points:

L93-95. This sentence sounds conclusive. I would suggest moving it to the Implications and use a more descriptive expression here.

L153: It is not clear which period of the 1 Sv decline is referring to. In Fig. 4a, the curves apparently reach peak in the mid 1990s, and have declines since then. By 2020 the decline is 5-6 Sv for the FULL and 4 Sv for the SENS.

L160: Please break this long sentence into short sentences. In addition, please check through the text and try to use shorter sentences/simplified expressions and improve clarity. Even though it was a pleasant read, I had to stop frequently as the texts are sometimes difficult to comprehend. Another example is the sentence L202-206.

L199. Continental boundaries -> continental slopes. "Boundaries" have multiple meanings and could be misleading.

L203: It would be nice to have the key locations marked in Fig. 1, e.g., Cape Farewell, Denmark Strait, Newfoundland Basin (maybe).

L312: Add a comma after "(Fig. 5b)".

L490: Please spell out c.i.

Fig. 5: I would suggest adding another map plot similar to Fig. 5a, but for the SENS experiment. This map would make the conclusions less speculative if the SENS map is able to show appreciably differences, notably in the Irminger Sea area. In addition, I find the results at sites 4 and 5 are less convincing compared with the other three sites. I would suggest adding the SENS experiment

results for cites 2-5 too as dashed, colored lines.

Reviewer #2 (Remarks to the Author):

Review

The paper by Böning and coauthors titled 'Decadal changes in Atlantic overturning due to the excessive 1990s Labrador Sea convection' presents an interesting contribution to the scientific quest of understanding the role of the Labrador Sea convection for low frequency AMOC variability.

Individual findings include 1) fast spreading of Labrador Sea Water and 2) downwelling in the boundary currents linked to eddy activity, neither of which is new. For example, 2) it has long been argued that sinking of dense waters that constitute the downwelling limb of the AMOC must take place near continental boundaries. In regions of strong eddy activity (near continental boundaries) nonlinear eddy fluxes of vorticity play an important role in balancing the vortex stretching associated with the sinking (e.g. Ref 31 and citing literature).

The novelty of the manuscript rests in pointing to the combined effects by which density (thickness) anomalies generated in the Labrador Sea enter the deep western boundary currents and impact the AMOC strength. Two critical aspects or criteria's are discussed; 1) formation of exceptionally dense Labrador Sea Water and/or 2) anomalies with decadal time scales. The manuscript tends to emphasize the former (1) as the novel result, but seems undecided (e.g. line 212-214). From the experimental design it is likely impossible to differentiate the two. In fact the latter (2) has recently been addressed: Yeager et al. 2021 (ref 41) showed that layer thickness anomalies are mainly important for AMOC dynamics on long (multidecadal) timescales.

The manuscript explores a single forced ocean-only simulation at high eddy resolving resolution with very appealing characteristics, complemented again by a single realization of sensitivity experiment excluding surface heat flux feedbacks over the convective regions of the Labrador Sea, in turn limiting the formation of Labrador Sea Water. The sensitivity experiment mainly serves the purpose of establishing a plausible range of AMOC impacts. For such an experimental design, the standard would be that uncertainty (robustness) is addressed (with limitations) by an ensemble approach. This is not the case here and the manuscript offers no uncertainty estimates on any results. This is a major limitation.

A revised manuscript should more clearly argue on the novelty at process level, quantify processes and present uncertainties.

Minor comments and recommendations:

Critical temperature/salinity relationship for Labrador sea water impact on overturning is jeopardized in the experiment SENS. This needs a note.

A significant part of the paper is dedicated to presenting and evaluating the VIKING20X without introducing new concepts and data, despite the fact that the specific configuration has been published on, also addressing related topics.

The manuscript could more clearly address the fact that 26N data indicate that the contribution from the 'sill' explains key aspects of the changes observed.

Conclusions appear to be rather delicately tied to the analysis of layer thickness anomalies, assuming this quantity has passive tracer properties disregarding wave propagation/dynamics.

Line 21: Consider if the use of the term 'a prominent repercussion' can be justified.

Line 22: Data is not presented to support the statement "unique period of excessive winter cooling". This could be considered.

Line 72-75: It is relevant to explain that the climatic trends are removed regionally and to discuss why this may not affect the dynamics at the periphery of the area (Methods). Likewise, it can be problematic to decouple wind stress and ocean induced mixing from heat fluxes, heat from salinity fluxes – if that is indeed how the experiment is designed?

Line 82: Consider a definition, also relevant to introduce in the context of previous discussions.

Line 91: Only heat fluxes are manipulated, not the atmospheric forcing in general.

Line 93: The paragraph here does not discuss or present more than one case, the CTRL. Consider rewording, what follows is an evaluation of the CTRL AMOC.

Line 116: 'northeastern basins' is possibly not sufficiently well defined.

Line 117-121. It seems direct to conclude that if western portion cannot be estimated meaningfully due to density compensation, then subtracting East and Full does not resolve the issue.

Line 122: The period 2014-2018 is not the one used in Figure 4c, this can be improved.

Line 124-127: Is a negligible contribution expected? Discuss how results here can be compared with the 2.6 Sv established in Ref 18.

Line 132-134: 26N would support a process modifying the LNADW, but this is not what is proposed here?

Line 135: Consider the use of notation.

Line 158-159: Here and above conclusions are indirect. Consider reducing and revising.

Line 169-164: Explain 'the prominent transport enhancement'?

Line 164: This is simply relying on the experimental design and looking at the lack of a 1998 peak, correct? Consider a shorter paragraph; there is little new information here.

Line 167-176/176-183: Descriptive/model evaluation, established knowledge, consider shortening.

Line 177-186: Fig. 5b,c,d shows the layer thickness. Consider not using "density peak".

Line 186-187: It is stated that the density peak (layer thickness anomaly) appears earlier at pos 5 than at pos 4, this is not clear from 5d.

Line 187: It is not clear from Fig 5d that the signal is more pronounced at pos 5.

Line 187-189: This suggestion is not well substantiated. Explain also how this relates to the main findings (line 26-27).

Line 191-192: Comparing Fig 4a with Fig 5d/pos 5? This is not obvious and appear to be a critical observation.

192-212: There is no reference to specific findings in the present study.

212-214: Wording here reflects a relatively modest contribution to the scientific literature reviewed above and not well quantified (time-scales, entrainment). Consider the relevance of the paragraph.

Line 223-224: Duration of the anomaly cannot be ruled out as the critical factor based on the analysis and experimental design. Rewording would be needed. Otherwise a convincing paragraph.

Line 230: It is the density of the lower north Atlantic deep water that is being modified, not the density of the overflow, consider specifying.

Line 230-232: Discuss how this fit (in time) with the transition in the 26N AMOC record, attributed to LNADW changes.

Figure 1: Include the date (season) of the snapshot.

Figure 1: Indicate where heat fluxes are fixed for the SENS experiment – outside the BC's but not exactly the grey area, correct?

Figure 1: Indicate the area used for Figure 2a. Explain why it is not the area defined in Methods?

Figure 4: Include SENS for 48N.

Line 505: Elaborate on '... is similar to Full at 48N', I expect it somehow relates to SENS vs CTRL?

Line 571: b) is anomalies.

Line 630-632: Identifying a pathway is not trivial, visual guidance needed.

Reviewer #3 (Remarks to the Author):

The manuscript 'Decadal changes in Atlantic overturning due to the excessive 1990s Labrador Sea convection' is clear, concise, interesting, and provides a novel approach to a recent topic: does the convection within the Labrador Sea impact the AMOC? The authors carried out two numerical simulations using a global model with a high resolution nest within the Atlantic. One simulation used relatively mild atmospheric forcing over a lengthy period of time such that the simulations could be examined on their surface forcing and AMOC production. This not only helped the authors address if the strong convection in the 1990s influenced the AMOC, but also what a Labrador Sea with mild forcing results with.

The authors find that, in general, Labrador Sea convection does influence AMOC transport slightly, but is dwarfed by what occurs upstream in the northeastern Atlantic. Forcing a large reduction in LSW convection (via their SENS simulation with weak atmospheric forcing) only reduced the total AMOC transport by a small amount. This conclusion isn't novel on its own, but this is the first time I've seen it shown in this manner, further showing that Labrador Sea convection isn't a dominant participant in total AMOC transports

I'm convinced that LSW production doesn't add much AMOC transport (10-20%) on top of the other regions, but I'm not convinced that LSW production isn't crucial to the AMOC itself (the authors do not argue this, nor do I recommend it). I have a few questions:

-The Labrador Sea releases a lot of heat to produce its water mass. While you show that this water mass doesn't have a large roll in AMOC transport, I wonder if that is because the Labrador Sea is the last convection region along the DWBC. I wonder how strongly these conclusions regarding LSW production and AMOC hold up if we numerically shut down convection upstream. This is clearly a separate study, I'm just hoping if the authors could discuss this a bit more.

-There are multiple papers out there that show if you shut down convection in the Labrador Sea (via Greenland hosing experiments for example), the AMOC eventually reduces much more than SENS shows here. While those hosing experiments would also impact the NE Atlantic to some extent, discussion on why we see larger AMOC changes in those experiments and not in SENS could be interesting.

I have very few minor/technical comments- the paper reads great:

L280-281: Can you elaborate why the May 1980-April 1981 window was chosen

L303-305: This sentence is a bit awkward and could use some polish

And answering the questions given to me by the reviewing template:

-The work supports the authors conclusions and claims. Extra evidence would be nice but isn't crucial for this manuscript

-I see no flaws in data analysis, interpretation and/or conclusions. My comments above would be nice to see addressed but they are not critical for publication

-Their methodology is sound, we have seen plenty of papers using a similar sensitivity simulation approach to investigate certain regions and their influence on ocean dynamics. Out of curiosity, I would have liked to see if a global/regional 1/4 or 1/12 simulation with similar forcing would have similar results.

-I could reproduce the results of this manuscript given their methodology

RESPONSE TO REVIEWER COMMENTS

Reviewer #1

This study explores the role of the Labrador Sea in affecting the inter-decadal variability of the AMOC with a high-resolution nested model. They find that by masking out the strong changes in heat fluxes of 1990s over the Lab Sea than therefore the Labrador Sea Water (LSW) production, the AMOC inter-decadal variability is affected upstream over the Irminger Sea instead of downstream over the Newfoundland Basin. The explanation they provide is that the anomalous dense Labrador Sea Water (LSW) during the 1990s invade northeastward into the Irminger Sea and get entrained into the AMOC lower limb. I found this study overall very interesting in three aspects. First, it works on a hot topic; no consensus on whether and how the Lab Sea plays a role in the AMOC variability has emerged since the OSNAP observation. Second, the results clearly show that “what happens in the Lab Sea stays in the Lab Sea” does not occur in their model; instead, the signal transits to the Irminger Sea and affects the EAST side of the OSNAP section. Finally, their explanation is reasonable and seems to be supported by observational evidence (where I have some comments as listed below). I would recommend publication of the research. I have one concern that requires some clarification and a few additional minor points.

We thank the reviewer for the thoughtful comments and constructive questions.

The authors claim that the advective signal of the LSW at 1500 m is able to travel northeastward into the Irminger Sea and later get entrained into the AMOC lower limb. Meanwhile, they find the WEST has no significant difference while the EAST has a ~2 Sv difference between the CTRL and SENS (Fig. 4). One thing that is not clear to me, is why the export of the LSW during the peak period is not detected along WEST, but its invasion to the Irminger Sea and entrainment to the lower limb could be easily detected at EAST.

In the statement “WEST has no significant difference” we sense a possible misunderstanding, perhaps because our discussion of Fig. 4 has been too brief. The key point is that our approach for deriving the contribution to the AMOC by downwelling in the Labrador Sea is not based on the overturning at WEST, but strictly on the difference between FULL and EAST. Accordingly, a transport profile for WEST was shown only for one period (2014-18) in Extended Figure 2 (now: Supplementary Figure 4) to emphasise that the distribution and partial compensation of north-south transports in density space (which had been noted in OSNAP studies before) implies that it cannot be used in a simple way for estimating the contribution of the downwelling in Labrador Sea to the AMOC. Accordingly, our finding of a rather small, ~1 Sv, additional contribution of the Labrador Sea to the AMOC peak in the 1990s does not mean that the strong LSW anomaly has left no mark in the individual overturning at WEST: this is now explicitly shown in the supplementary figure by including the period 1995-1999 in Supplementary Figure 4. - In the revised manuscript, we attempt to clarify this point by significantly expanding the discussion of Fig. 4, and by including the response of the WEST, EAST and FULL transport profiles to the LSW peak period in the supplementary figure.

In addition, if the signal is advective as shown in the model, should its northward transport across EAST and later transport back cancel out, as both directions are within the lower limb? Am I missing something? Some clarification is expected here.

It is true that the northward advection of LSW across EAST occurs in a density range well within the lower, southward limb of the AMOC. However, what needs to be considered also is the strong horizontal variation of the lower limb transport (as discussed, e.g., by Lozier et al. (2019) and nicely illustrated in their Fig. 2): as a manifestation of the gyre circulation the lower limb flow is northward over most of the Irminger Basin (and over the eastern Iceland Basin), while the southward flow is concentrated in the western boundary current.

Minor points:

L93-95. This sentence sounds conclusive. I would suggest moving it to the Implications and use a more descriptive expression here.

We have re-formulated the sentence.

L153: It is not clear which period of the 1 Sv decline is referring to. In Fig. 4a, the curves apparently reach peak in the mid 1990s, and have declines since then. By 2020 the decline is 5-6 Sv for the FULL and 4 Sv for the SENS.

The “~1 Sv” should refer to the difference between CTRL and SENS in the strength of the decline. We agree that the statement was not entirely clear. In order to improve the clarity, parts of the paragraph have been revised.

L160: Please break this long sentence into short sentences. In addition, please check through the text and try to use shorter sentences/simplified expressions and improve clarity. Even though it was a pleasant read, I had to stop frequently as the texts are sometimes difficult to comprehend. Another example is the sentence L202-206.

This and other sentences/expressions throughout the manuscript have been re-written.

L199. Continental boundaries -> continental slopes. “Boundaries” have multiple meanings and could be misleading.

Done

L203: It would be nice to have the key locations marked in Fig. 1, e.g., Cape Farewell, Denmark Strait, Newfoundland Basin (maybe).

Figure 1 has been edited

L312: Add a comma after “(Fig. 5b)”.

Done

L490: Please spell out c.i.

Done

Fig. 5: I would suggest adding another map plot similar to Fig. 5a, but for the SENS experiment. This map would make the conclusions less speculative if the SENS map is able to show appreciable differences, notably in the Irminger Sea area. In addition, I find the results at sites 4 and 5 are less convincing compared with the other three sites. I would suggest adding the SENS experiment results for sites 2-5 too as dashed, colored lines.

We followed the suggestion and complemented Fig. 5 with the respective results for SENS; in addition, the prior Ext. Data Fig. 4 was edited and moved to become Fig. 5c. Because of the additional panels, the figure was split into two: Fig. 5 and Fig. 6. (In addition, Position 5 was slightly shifted, so that it now corresponds exactly to the mooring site M1 of the OSNAP programme.)

Reviewer #2

The paper by Böning and coauthors titled 'Decadal changes in Atlantic overturning due to the excessive 1990s Labrador Sea convection' presents an interesting contribution to the scientific quest of understanding the role of the Labrador Sea convection for low frequency AMOC variability.

We thank the reviewer for the thoughtful comments and constructive questions.

Individual findings include 1) fast spreading of Labrador Sea Water and 2) downwelling in the boundary currents linked to eddy activity neither of which is new. For example, 2) it has long been argued that sinking of dense waters that constitute the downwelling limb of the AMOC must take place near continental boundaries. In regions of strong eddy activity (near continental boundaries) nonlinear eddy fluxes of vorticity play an important role in balancing the vortex stretching associated with the sinking (e.g. Ref 31 and citing literature).

The novelty of the manuscript rests in pointing to the combined effects by which density (thickness) anomalies generated in the Labrador Sea enter the deep western boundary currents and impact the AMOC strength. Two critical aspects or criteria's are discussed; 1) formation of exceptionally dense Labrador Sea Water and/or 2) anomalies with decadal time scales. The manuscript tends to emphasize the former (1) as the novel result, but seems undecided (e.g. line 212-214). From the experimental design it is likely impossible to differentiate the two. In fact the latter (2) has recently been addressed: Yeager et al. 2021 (ref 41) showed that layer thickness anomalies are mainly important for AMOC dynamics on long (multidecadal) timescales.

The manuscript explores a single forced ocean-only simulation at high eddy resolving resolution with very appealing characteristics, complemented again by a single realization of

sensitivity experiment excluding surface heat flux feedbacks over the convective regions of the Labrador Sea, in turn limiting the formation of Labrador Sea Water. The sensitivity experiment mainly serve the purpose of establishing a plausible range of AMOC impacts. For such an experimental design, the standard would be that uncertainty (robustness) is addressed (with limitations) by an ensemble approach. This is not the case here and the manuscript offers no uncertainty estimates on any results. This is a major limitation.

A revised manuscript should more clearly argue on the novelty at process level, quantify processes and present uncertainties.

We would like to stress that the novelty of our findings should not be seen at the process level (e.g., the invasion of LSW anomalies into the Irminger Sea and its entrainment in the deep wbc, etc.), but in demonstrating a mechanism of how the AMOC can get affected by (inter-)decadal changes in LSW upstream, over the Irminger Sea, instead of downstream over the Newfoundland Basin. We believe that exposing this remote effect can advance the current debate on the role of the Labrador Sea, and may represent an important aspect to consider in the planning of future monitoring programmes.

In order to provide some assessment of the robustness of the decadal variation in LSW formation and its impact on the AMOC we have included in the revised manuscript brief discussions (with several supplementary figures) of three additional hindcast simulations in VIKING20X: they differ from CTRL in the atmospheric forcing product (CORE versus JRA55-do), the freshwater forcing (which may be regarded as the most uncertain component of subpolar forcing), and/or an initialisation from different states in 1980 instead of 1958; the supplementary experiments are described in Methods and summarised in Supplementary Table 1. We show that in this series there is little sensitivity in the key model behaviours relevant for the main conclusions; in particular, the intriguing correspondence between the inter-decadal changes of the AMOC and the abyssal density of the wbc in the Irminger Sea noted in CTRL and SENS holds in these experiments as well.

We think that more comprehensive investigations of model sensitivities and uncertainties would require a multi-model intercomparison effort - a daunting task given the very large requirements for performing model simulations with the very fine mesh sizes needed for capturing the critical processes governing the dynamics in the subpolar North Atlantic. We argue (final paragraph of Discussion), based on previous studies, that a main uncertainty of the LSW-AMOC relations obtained in model simulations lies in the realistic representation of the density difference between the overflow and the LSW: while this is significantly improved in the present configuration thanks to the very high resolution (cf. references 19,20,22), a reliable quantitative estimation of uncertainty due to such critical factors cannot be given without dedicated further experimentation efforts.

Minor comments and recommendations:

Critical temperature/salinity relationship for Labrador sea water impact on overturning is jeopardised in the experiment SENS. This needs a note.

We do not regard the impact as “jeopardised”. The atmospheric state prescribed in SENS is not an unnatural invention, but based on a real year with moderate air-sea fluxes: as shown in Fig. 2, these result in a convection intensity and LSW properties rather typical for the periods before and after the exceptional 1990s’ phase. We think that Fig. 2 convincingly demonstrates that the intention of SENS - to test the hypothesis that the increase of the AMOC simulated in CTRL was caused by the excessive cooling during that period - is well achieved with this set-up, since it succeeded in curbing the formation of the dense vintage of LSW, while retaining the formation of the lighter LSW continuously formed in winters of weak to moderate convection intensity. Since the interest here is solely on the dynamical effect of this event, we think it would be superfluous to supplement the presentation of the density changes with a discussion of the concomitant modifications in T/S-relationships.

A significant part of the paper is dedicated to presenting and evaluating the VIKING20X without introducing new concepts and data, despite the fact that the specific configuration has been published on, also addressing related topics.

We regard it as an important asset of our study that it rests on a model configuration whose behaviour with respect to individual processes critical for the dynamics of the subpolar North Atlantic, has been thoroughly assessed in recent publications: e.g., eddy variability (ref 20), convection variability (ref 22) and DWBC (ref 16) in the Labrador Sea or the maintenance of the outflow plume (ref 19). It is because of these studies that we feel confident to use this model for addressing the hot and contentious topic of whether and how LSW formation can affect the AMOC in the subpolar North Atlantic.

The manuscript could more clearly address the fact that 26N data indicate that the contribution from the ‘sill’ explains key aspects of the changes observed.

We agree that this is an important issue, and we have added it more explicitly in the revised manuscript. In the Introduction we included the observation (ref 9) that the AMOC-decline at 26N was confined to the lower NADW layer, without changes in the LSW layer. To our knowledge, there is no convincing explanation yet of the origin of this deep trend, since there has been no significant trend observed in DSOW, the primary source water of LNADW. We take this up in the Discussion, arguing that the mechanism of the inter-decadal AMOC changes suggested in our study does offer a possible explanation, via the mixing/entrainment of the strong LSW anomalies into the wbc between Denmark Strait and Cape Farewell.

Conclusions appear to be rather delicately tied to the analysis of layer thickness anomalies, assuming this quantity has passive tracer properties disregarding wave propagation/dynamics.

The strong changes in the thickness of the LSW layer represents a prime characteristic of the 1990s-period. Because this signal is so large, it appears ideal for the present purpose of illustrating the spreading of the LSW anomaly. We believe that on the short time scales involved in the Labrador Sea - Irminger Sea connection, the

spreading (in the interior basins) is sufficiently explained as a primarily advective mechanism, without invoking wave processes (which, on these time scales, might be more relevant along the continental slopes). We therefore choose to not complicate matters by entering into a discussion of details in the dynamics of the spreading, believing that this would rather distract from the main points of the paper.

Line 21: Consider if the use of the term 'a prominent repercussion' can be justified.

The Abstract has been substantially revised, the term is no longer used here.

Line 22: Data is not presented to support the statement "unique period of excessive winter cooling". This could be considered.

In the revised Abstract the statement is reduced to "The exceptionally cold winters". (Note that in the studies of the hydrographic changes in the Labrador Sea referenced in the paper, the 1990s-period is described as the one with the "coldest, densest, deepest and most voluminous LSW since the 1930s", with commonly used characterizations such as "extremely deep convection", "excessive convection" with generation of "exceptional LSW".)

Line 72-75: It is relevant to explain that the climatic trends are removed regionally and to discuss why this may not affect the dynamics at the periphery of the area (Methods). Likewise, it can be problematic to decouple wind stress and ocean induced mixing from heat fluxes, heat from salinity fluxes – if that is indeed how the experiment is designed?

The technique of performing sensitivity experiments in which the variability of certain fluxes (e.g., of momentum, freshwater or heat) is selectively suppressed has successfully been used in several studies (e.g., references to five North Atlantic applications are given in ref 7); this includes applications where subcomponents of the atmospheric state variability are selectively changed in certain regions (e.g., in the Labrador Sea). We believe that in this regard, SENS is based on a technique that is sufficiently established in ocean modelling so that an explanation of technical details which have all been discussed in the literature would be out of place here.

Line 82: Consider a definition, also relevant to introduce in the context of previous discussions.

We think that in the general context used here (as in the previous paragraphs), it is sufficient to introduce "Labrador Sea Water" as the water mass which is "convectively formed in the Labrador Sea" - as done in the present sentence. In order to avoid cluttering the main text with details (which become important for the quantitative elaborations later on), we chose to defer the definition of the range of the "dense" LSW, used e.g. for Figs 5 and 6, to Methods.

Line 91: Only heat fluxes are manipulated, not the atmospheric forcing in general.

The sentence has been reworded.

Line 93: The paragraph here does not discuss or present more than one case, the CTRL. Consider rewording, what follows is an evaluation of the CTRL AMOC.

The sentence has been removed.

Line 116: 'northeastern basins' is possibly not sufficiently well defined.

We think that it must be pretty clear from the context that the 'northeastern basins' refer to 'all areas of the subpolar North Atlantic north of the EAST section' - we prefer to keep it simple, so as not to distract from the fundamental question posed in the sentence.

Line 117-121. It seems direct to conclude that if western portion cannot be estimated meaningfully due to density compensation, then subtracting East and Full does not resolve the issue.

It seems that the statement was not sufficiently explained; we have expanded it for clarification. One could of course use the overturning transport at WEST to calculate the contribution of downwelling in the Labrador Sea to the AMOC. But this contribution is not simply given by its bulk value of overturning: one would need to account for the different streamfunction profiles in density space in EAST and WEST and compute their sum for each density bin separately. As illustrated by Supplementary Fig. 4, the net result of the bin-by-bin addition is a streamfunction profile at FULL whose maximum is shifted in density space compared to EAST. One would need to go this way if one wanted to give a comprehensive account of the three-dimensional (lat - lon - density) evolution of the overturning in the subpolar North Atlantic - but this is not pertinent to the objectives of the present study: of interest here is only the net contribution of the downwelling north of WEST to the evolution of the AMOC from EAST to FULL, and this is simply and most directly given by the difference of their streamfunction maxima. - We have expanded the statement for clarification (in the main text and Supplementary Figure 4).

Line 122: The period 2014-2018 is not the one used in Figure 4c, this can be improved.

The period was used in the computation, but incorrectly stated in Fig. 4c - we corrected this.

Line 124-127: Is a negligible contribution expected? Discuss how results here can be compared with the 2.6 Sv established in Ref 18.

The gain between EAST and FULL discussed here must not be compared with the 2.6 Sv obtained for WEST by Li et al (see also the previous comments re. line 117-121), but to the corresponding difference between FULL and EAST in the observations (which is nil). We have added a respective explanation.

Line 132-134: 26N would support a process modifying the LNADW, but this is not what is proposed here?

We refer to our response given above to the reviewer's suggestion of discussing the changes in the LNADW transport at 26N. - We think that the discussion of this important point (i.e., proposing these changes to be a repercussion of changing LSW properties in the entrainment, and not in the overflow itself) should not be given here but in the Discussion section.

Line 135: Consider the use of notation.

The whole paragraph has been revised to improve clarity.

Line 158-159: Here and above conclusions are indirect. Consider reducing and revising.

The whole paragraph has been revised to improve clarity.

Line 169-164: Explain ‘the prominent transport enhancement’?

The sentence has been revised.

Line 164: This is simply relying on the experimental design and looking at the lack of a 1998 peak, correct? Consider a shorter paragraph; there is little new information here.

As above

Line 167-176/176-183: Descriptive/model evaluation, established knowledge, consider shortening.

The northeastward spreading of the 1990s’ LSW anomaly into the Irminger Sea and its manifestation in the western boundary current is central to the conclusions of this paper. We think it is important to note that, while that pathway has been established in observations (and in that sense there is indeed little new information here), its potential relevance to the debate of whether and how LSW formation can contribute to the AMOC has not been considered yet. We believe that for stimulating that discussion on the basis of model results, a sufficiently detailed description of the simulated spreading behaviour is absolutely essential. - Note that in this regard (also in response to other comments) we have expanded Fig. 5 (and separated it in two parts, Fig. 5 and Fig. 6).

Line 177-186: Fig. 5b,c,d shows the layer thickness. Consider not using “density peak”.

We have substituted “density anomaly” by “LSW layer thickness anomaly”, and “density peak” by “LSW peak”.

Line 186-187: It is stated that the density peak (layer thickness anomaly) appears earlier at pos 5 than at pos 4, this is not clear from 5d.

There have been some changes to Fig. 5 in our revision (partly in response to other reviewers’ suggestions, e.g., to include the respective behaviour of SENS). As part of the revision of the time series (now: Fig.6), we chose Pos. 5 to correspond more exactly to the mooring position M1 of the OSNAP programme. Statements in the text have been adapted accordingly.

Line 187: It is not clear from Fig 5d that the signal is more pronounced at pos 5.

The statement was incorrect and has been revised (as part of the changes noted above).

Line 187-189: This suggestion is not well substantiated. Explain also how this relates to the main findings (line 26-27).

The statement that part of the LSW may enter the wbc via a shortcut in the Southern Irminger Sea is based on the respective current pattern in Fig. 5c (now edited and moved here from Ext. Data). This is certainly not enough for drawing any quantitative conclusions about the relevance of this pathway. Since the feature is just a mere detail in the broader picture of the LSW spreading to the Irminger wbc, with no consequences for the main findings, we have removed it in the revision.

Line 191-192: Comparing Fig 4a with Fig 5d/pos 5? This is not obvious and appear to be a critical observation.

We have substantiated the suggestion of a correspondence between the (inter-decadal) changes of the overturning and the density evolution in the deep boundary current by including an additional figure for CTRL and SENS (Fig. 7), supplemented by analogous plots for the additional hindcasting experiments (Suppl. Fig. 7c,d). The paragraph has been revised to clear up the context for the discussion of this potentially important feature.

192-212: There is no reference to specific findings in the present study.

The section has been completely revised (with additional figures), aiming at a clearer discussion of the present findings in the context of current understanding.

212-214: Wording here reflects a relatively modest contribution to the scientific literature reviewed above and not well quantified (time-scales, entrainment). Consider the relevance of the paragraph.

The paragraph has been completely revised.

Line 223-224: Duration of the anomaly cannot be ruled out as the critical factor based on the analysis and experimental design. Rewording would be needed. Otherwise a convincing paragraph.

Yes, we agree and have expanded the statement to clarify this point.

Line 230: It is the density of the lower north Atlantic deep water that is being modified, not the density of the overflow, consider specifying.

The statement has been clarified.

Line 230-232: Discuss how this fit (in time) with the transition in the 26N AMOC record, attributed to LNADW changes.

We elaborated on this point in the Discussion.

Figure 1: Include the date (season) of the snapshot.

Done

Figure 1: Indicate where heat fluxes are fixed for the SENS experiment – outside the BC's but not exactly the grey area, correct?

Figure 1 has been revised, including (among other pieces) a depiction of the area as suggested.

Figure 1: Indicate the area used for Figure 2a. Explain why it is not the area defined in Methods?

The area used for the computation of the mixed layer depth has been indicated in Figure 1; it is the same as defined in Methods. (The area is chosen to encompass the parts of the Labrador Sea with the deepest convection in every winter.)

Figure 4: Include SENS for 48N.

Since the difference between CTRL and SENS for 48N is similar to the respective difference for FULL, we think there is little new information here. In order to avoid a cluttering of Figure 4 with even more lines, we have chosen to not include this one, but to clarify the explanation in the caption.

Line 505: Elaborate on ‘... is similar to Full at 48N’, I expect it somehow relates to SENS vs CTRL?

Yes, the statement was meant to refer to the difference between CTRL and SENS, which at 48N is similar to FULL (see preceding comment). We changed the formulation to improve clarity.

Line 571: b) is anomalies.

Yes - we corrected this.

Line 630-632: Identifying a pathway is not trivial, visual guidance needed.

We followed this recommendation and illustrated the pathway by broad arrows. (The figure has become Fig. 5c.)

Reviewer #3

The manuscript 'Decadal changes in Atlantic overturning due to the excessive 1990s Labrador Sea convection' is clear, concise, interesting, and provides a novel approach to a recent topic: does the convection within the Labrador Sea impact the AMOC? The authors carried out two numerical simulations using a global model with a high resolution nest within the Atlantic. One simulation used relatively mild atmospheric forcing over a lengthy period of time such that the simulations could be examined on their surface forcing and AMOC production. This not only helped the authors address if the strong convection in the 1990s influenced the AMOC, but also what a Labrador Sea with mild forcing results with.

The authors find that, in general, Labrador Sea convection does influence AMOC transport slightly, but is dwarfed by what occurs upstream in the northeastern Atlantic. Forcing a large reduction in LSW convection (via their SENS simulation with weak atmospheric forcing) only reduced the total AMOC transport by a small amount. This conclusion isn't novel on its own,

but this is the first time I've seen it shown in this manner, further showing that Labrador Sea convection isn't a dominant participant in total AMOC transports

I'm convinced that LSW production doesn't add much AMOC transport (10-20%) on top of the other regions, but I'm not convinced that LSW production isn't crucial to the AMOC itself (the authors do not argue this, nor do I recommend it). I have a few questions:

We thank the reviewer for the thoughtful comments and constructive questions. We agree with the notion that a small regional contribution to the AMOC in the Labrador Sea during both weak and strong convection phases does not mean that the role of LSW production in the AMOC can be neglected altogether. In accordance with our experimental focus on the LSW-AMOC relation on (inter-)decadal timescales, we largely refrained from discussing possible implications for more general aspects such as the role of LSW production in the “mean” AMOC. The review has induced us to somewhat expand this discussion in the revision, guided by the questions asked. From the thoughts provided below it should become apparent, however, that a more thorough deliberation would be beyond the constraints of this paper - and perhaps of any single study on the dynamics of the subpolar North Atlantic.

-The Labrador Sea releases a lot of heat to produce its water mass. While you show that this water mass doesn't have a large roll in AMOC transport, I wonder if that is because the Labrador Sea is the last convection region along the DWBC. I wonder how strongly these conclusions regarding LSW production and AMOC hold up if we numerically shut down convection upstream. This is clearly a separate study, I'm just hoping if the authors could discuss this a bit more.

The Labrador Sea is not only the last convection region along the DWBC, it also produces the densest water in the subpolar North Atlantic (south of the ridges). In the study we find the increase in the density and volume of LSW in the 1990s affecting the AMOC via its export into the Irminger Sea and its entrainment in the DWBC. Assuming that the linkage between the boundary density and EAST transport found here holds more generally (a hypothesis that needs to be investigated in further study), we would expect the transport to be governed mainly by the density of two source water masses: the outflow from the Nordic Seas, and the ambient water entrained into the near-bottom plume. To the extent that the latter (i.e., the water directly overlying the plume) is provided primarily by the intermediate water with the highest density formed in the subpolar basins, it can be assumed that changes in the formation of more buoyant subpolar mode waters are of secondary importance.

Concerning water mass changes farther upstream, in the Nordic Seas, the situation is very different. There have been several studies documenting their leading role in (low-frequency) AMOC changes and trends. Pertinent examples include: the finding that anthropogenic AMOC trends in different climate simulations were controlled by the density of the intermediate waters in the Nordic Seas (Schweckendiek and Willebrand, J. Clim. 2005); and the finding in a large model ensemble of a tight linkage between multi-decadal AMOC trends and the Denmark Strait sill density (Latif et al., J. Clim. 2006). Regarding the relative role of changes in LSW production and changes in outflow density, it may be warranted to renew attention to an earlier suite of sensitivity experiments within the WOCE Community Modelling Effort (Döscher and

Redler, JPO 1997). Their key finding was that the effect of Labrador Sea convection on the AMOC strongly depends on the density of the outflow: the LSW influence is small only if the density of the outflow (after its mixing with the ambient water) is as high as in current observations; in contrast, the AMOC's sensitivity to convection changes can become much stronger if this density is smaller. - We note (with relevance also to the question addressed below) that the latter case is typical for most ocean climate models with an insufficient resolution of the downslope flow regime, which results in a too strong dilution of the outflow by spurious numerical mixing (Colombo et al., GMD 2020).

-There are multiple papers out there that show if you shut down convection in the Labrador Sea (via Greenland hosing experiments for example), the AMOC eventually reduces much more than SENS shows here. While those hosing experiments would also impact the NE Atlantic to some extent, discussion on why we see larger AMOC changes in those experiments and not in SENS could be interesting.

Discussing the present results on the role of the Labrador Sea in decadal AMOC variability in the context of hosing experiments is not straightforward. Key caveats are that in those studies freshwater flux anomalies were typically applied over a broad swath of the northern North Atlantic (a typical choice being 50°N-70°N) and the very large amplitude of the freshwater input. This not only shuts down Labrador Sea convection completely (whereas the forcing anomaly in SENS is comparatively mild, with moderate convection and renewal of upper LSW still taking place every winter), but also has a large, immediate effect on the water mass transformation in the Nordic Seas. As outlined above, it is thus likely that the response of the AMOC is eventually governed to a substantial degree by the latter, so that assessing the individual effect of the Labrador Sea convection becomes virtually impossible.

As a possible alternative it may be interesting to look at the growing number of studies directed at the impact of the increasing meltwater runoff from Greenland. Since in realistic settings an increasing runoff occurs mainly along the southeastern and western coasts of Greenland, the first convection area affected by the ensuing surface freshwater anomalies is the Labrador Sea. More specifically, there have been some simulations with eddy models addressing the response to a sudden large increase in the runoff which typically involved a rapid suppression of Labrador Sea convection (Weijer et al. GRL 2012; Den Toom et al. JPO 2014; Böning et al. Nature Geosci. 2016; Martin and Biastoch Ocean Sci 2023 - a reference to the latter study has been included). The AMOC response during the first 1-2 decades after the convection shut-off (i.e., before the freshwater anomalies had time to spread into the northeastern basins and the Nordic Seas) can be attributed primarily to the individual effect of the Labrador Sea. Its decline in these experiments is 4-5 Sv which is not much larger than the difference between the phases of weak and strong convection in CTRL.

I have very few minor/technical comments- the paper reads great:

L280-281: Can you elaborate why the May 1980-April 1981 window was chosen

We have added the following explanation and a supplementary figure to the Methods section:

The choice of the period 1st May 1980 - 30th April 1981 is based on two considerations. First, on the intention to selectively suppress the effect of the strong increase in convection activity during the 1980s and early 1990s with minimal distortion of other forcing elements impacting the evolution of the AMOC; second, on minimising disturbances involved in step function changes in the day-to-day variation in forcing. The first criterion appears optimally met by the years 1980-1981 representing the transition of the North Atlantic Oscillation index from a prolonged negative phase to the increasingly positive phase of the 1980s and 1990s. This is reflected by the evolution of key forcing elements in CTRL, specifically the heat fluxes and convection intensity, which for the winter of 1980-1981 were close to their multi-decadal means (Supplementary Fig. 1). For the second we followed the development of a 'Repeated Year Forcing' scheme for JRA55-do by Stewart et al. (2020) who found that the step from 1st May to 30th April provided the smoothest transition period possible.

L303-305: This sentence is a bit awkward and could use some polish

The sentence was re-formulated.

And answering the questions given to me by the reviewing template:

-The work supports the authors conclusions and claims. Extra evidence would be nice but isn't crucial for this manuscript

-I see no flaws in data analysis, interpretation and/or conclusions. My comments above would be nice to seen addressed but they are not critical for publication

-Their methodology is sound, we have seen plenty of papers using a similar sensitivity simulation approach to investigate certain regions and their influence on ocean dynamics. Out of curiosity, I would have liked to see if a global/regional 1/4 or 1/12 simulation with similar forcing would have similar results.

Although we have performed similar hindcast simulations with coarser model configurations, specifically with ORCA025, we think that entering into a discussion of resolution dependencies would be outside the scope of the present paper. There are two main issues with coarser grids in simulations of the subpolar North Atlantic: (i) The lack of eddy fluxes that are found critical for the preconditioning and restratification of convection in the Labrador Sea, typically resulting in a much too wide and deep convection area even during mild winters; and (ii) failure in maintaining the density of the DSOW due to excessive spurious mixing in the outflow regime. Since these deficiencies can have critical repercussions for how the LSW formation variability influences the AMOC, we agree with the reviewer that a documentation of the resolution dependencies in AMOC simulations is warranted. We think, however, it would not make sense to just complement parts of the 1/20°-

presentation with corresponding output from 1/4°-experiments, without an elaborate discussion that would need to touch on a score of questions involved in lower resolution simulations, such as dependencies on parameterisation choices. In the paper we have given references to pertinent studies regarding these resolution issues; in our revision we try to emphasise the point in the concluding remarks, in the hope to raise awareness of the need for further systematic investigations of these critical dependencies.

-I could reproduce the results of this manuscript given their methodology

End

REVIEWER COMMENTS

Reviewer #1 (Remarks to the Author):

Comments on the revised manuscript entitled "Decadal changes in Atlantic overturning due to the excessive 1990s Labrador Sea convection", by Böning et al.

I would like to thank the authors for taking my suggestions seriously and spending time working on the revision. I am very impressed with the revised manuscript. The authors have addressed all of my concerns and have made significant improvements to the manuscript. I believe that the revised manuscript is now ready for publication in Nature Communications.

I have one minor suggestion for the revised manuscript. In Supplementary Figure 4, I would suggest either merging the two subplots into one plot (by lengthening the x-axis) or adding another legend for subplot b. In addition, the caption reads "in experiment CTRL for the period 2014-2018", but the figure and the first line of the caption both indicate "2014-2019". Please be consistent.

In the main text, lines 48 and 57 are missing two periods at the end of the sentences.

Reviewer #2 (Remarks to the Author):

This is now a very inspiring manuscript to read, it includes insightful discussions on a number of key open issues in the nexus between AMOC related observations, ocean modelling and descriptive physical oceanography. Despite the limitations, the journal format offers it is complete and forward looking.

The revision of the manuscript accommodates adequately many specific questions raised, and rewriting paragraphs has significantly helped in shedding light on the limitations of the approach and in reducing the risk of misunderstandings.

As emphasized by the authors in the response, the manuscript suggests (among other points) a possible mechanism connecting AMOC and LSW variability. The experimental design (SENS) supporting the discussion is simple with elegant features. It achieves total artificial suppression of LSW densification locally without apparent strong drift or chock effects. This suffice to demonstrate a mechanism, to discuss causal relations but is far too limited to make any quantitative statements, to address robustness and significance levels. This is still my main concern. It does not meet best practices in the field.

I may not have been sufficiently explicit on the need for assessing the sensitivity of the response seen in the SENS experiment. I apologize if that is the case. The main conclusions (AMOC impact) hinges on the subtle differences between CTRL and SENS in the latitude-time-density-overturning space. The storyline is very plausible, but the manuscript would strongly benefit from some quantification of uncertainty of the AMOC response. This would be possible through variations of SENS type nudged simulations. In the response letter the authors reject further simulations arguing that a multi-model approach is out of scope. I totally agree, but variations of SENS would not be. The authors have elegantly navigated around this point supplying a set of alternative CTRL simulations, which are useful but not sufficient. I find it likely that a result in the ballpark of the proposed 20% AMOC anomaly can be substantiated with some effort (variations of SENS simulations) but also conclude that this is not a path the authors are confident in pursuing. By further insisting on this, I risk delaying significantly the publication of an important and inspiring contributing to the ongoing debate. Instead, I propose

- 1) to remove completely the quantitative statements in the abstract and
- 2) to include a short paragraph acknowledging that the AMOC evolution seen in the SENS experiment is intimately linked to the experimental design and will vary with relaxed assumptions such as repeat year, initial year, domain etc.

A few minor comments:

Line 82: "Year to year variations in buoyancy forcing inhibited" – I am not totally convinced this is actually what is implemented (CORE II) even though this was the purpose. Some terms (e.g. evaporation) are likely still calculated internally from bulk formulations resulting in variations in year-to-year buoyancy forcing.

Line 288: reference missing I suspect.

Reviewer #3 (Remarks to the Author):

I'm happy with the author's revised version of their manuscript, both in terms of my earlier comments as well as the other reviewers. I have no further comments.

RESPONSE TO REVIEWER COMMENTS

We would like to express our sincere thanks to all three reviewers for their time and efforts spent with our submission – their comments and suggestions have been of great help for us in identifying issues in the materials and in improving the clarity of the presentation.

Reviewer #1

I would like to thank the authors for taking my suggestions seriously and spending time working on the revision. I am very impressed with the revised manuscript. The authors have addressed all of my concerns and have made significant improvements to the manuscript. I believe that the revised manuscript is now ready for publication in Nature Communications.

I have one minor suggestion for the revised manuscript. In Supplementary Figure 4, I would suggest either merging the two subplots into one plot (by lengthening the x-axis) or adding another legend for subplot b. In addition, the caption reads "in experiment CTRL for the period 2014-2018", but the figure and the first line of the caption both indicate "2014-2019". Please be consistent.

We have chosen to retain the two subplots for visual clarity but added another legend for panel b.

The period presented is 2014-2019 - we corrected the error in the caption.

In the main text, lines 48 and 57 are missing two periods at the end of the sentences.

Corrected (as well as some other typos).

Reviewer #2

This is now a very inspiring manuscript to read, it includes insightful discussions on a number of key open issues in the nexus between AMOC related observations, ocean modelling and descriptive physical oceanography. Despite the limitations, the journal format offers it is complete and forward looking.

The revision of the manuscript accommodates adequately many specific questions raised, and rewriting paragraphs has significantly helped in shedding light on the limitations of the approach and in reducing the risk of misunderstandings.

As emphasized by the authors in the response, the manuscript suggests (among other points) a possible mechanism connecting AMOC and LSW variability. The experimental design (SENS) supporting the discussion is simple with elegant features. It achieves total artificial suppression of LSW densification locally without apparent strong drift or chock effects. This suffice to demonstrate a mechanism, to discuss causal relations but is far too limited to make any quantitative statements, to address robustness and significance levels. This is still my main concern. It does not meet best practices in the field.

I may not have been sufficiently explicit on the need for assessing the sensitivity of the response seen in the SENS experiment. I apologize if that is the case. The main conclusions (AMOC impact) hinges on the subtle differences between CTRL and SENS in the latitude-time-density-overturning space. The storyline is very plausible, but the manuscript would strongly benefit from some quantification of uncertainty of the AMOC response. This would be possible through variations of SENS type nudged simulations. In the response letter the authors reject further simulations arguing that a multi-model approach is out of scope. I totally agree, but variations of SENS would not be. The authors have elegantly navigated around this point supplying a set of alternative CTRL simulations, which are useful but not sufficient. I find it likely that a result in the ballpark of the proposed 20% AMOC anomaly can be substantiated with some effort (variations of SENS simulations) but also conclude that this is not a path the authors are confident in pursuing. By further insisting on this, I risk delaying significantly the publication of an important and inspiring contributing to the ongoing debate.

It seems that the basis of our diverging views on SENS lies in a somewhat different perception of what this experiment is meant to achieve. In our concept it (*only*) serves to demonstrate that the interdecadal AMOC peak in the 1990s simulated in the control experiment(s) (as in many other previous model studies) is causally related to the formation of the exceptionally dense LSW. We believe that the configuration of SENS clearly succeeds in achieving this objective: by preventing the formation of this LSW variety the AMOC peak is eliminated. Beyond establishing this linkage (and helping to support the crucial role of the spreading of the LSW into the Irminger Sea as part of the AMOC response mechanism), we do not see further functions of SENS in the context of this study. In particular, in apparent contrast to the reviewer's view, we do not think that variations of the SENS set-up could furnish much further quantitative insights, e.g., to questions of robustness etc. The AMOC increase from the early-80s to the mid-90s is 20%-25% in all control configurations; this trend is removed in the current set-up of SENS. It is conceivable that by choosing a different initial year for the repeated forcing, e.g., later in the 1980s, or by geographically restricting the area for the heat flux anomaly, one could achieve any partial removal of the inter-decadal trend: in this way, one might be able to further refine the attribution of the trend to subregions of the Labrador Sea or limited forcing periods. We regard this, however, as a futile (though very expensive*) exercise in the context of the present study, since it would contribute almost nothing to the conclusions of the paper – especially it could not advance assessments of uncertainty. In that regard our take from previous modelling work is (as argued in the final paragraph of Discussion) that a main uncertainty of the LSW-AMOC relations lies in the simulation of the very small-scale

(partly, subgrid-scale) processes which, e.g., affect the realistic representation of the density difference between the overflow and the LSW: while this is significantly improved in the present model thanks to its very high resolution, a reliable quantitative estimation of uncertainty due to such factors cannot be given without a dedicated multi-model programme of experimentation - which one may hope is going to be stimulated with the present study.

*Due to the very high resolution a VIKING20X-simulation requires a significant share of the available cpu capacity on a high-performance computing (HPC) system: e.g., the SENS experiment ran continuously for 12 weeks on one of the fastest German HPC systems available to the scientific community, with constant surveillance by 1-2 scientists/scientific programmers.

Instead, I propose

1) to remove completely the quantitative statements in the abstract

The single statement with a (semi-)quantitative character is about the magnitude of the AMOC anomaly during the 1990s: the increase from the early 1980s is robustly between 20% and 25% in all control experiments. As demonstrated by SENS, the trend can be attributed to the excessive cooling in the Labrador Sea. As argued above, with different set-ups of SENS one might get some refined insights into the relative contribution of various sub-regions in the Labrador Sea, but nothing to tackle the statement that “the exceptionally cold winters in the Labrador Sea during the first half of the 1990s induced a positive AMOC anomaly of more than 20%”. We believe that this statement gives a good summary of the findings, and we would propose to keep it essentially as is – including the “20%” which we believe serves as a useful up-front information for the reader about the order-of-magnitude of the phenomena investigated in the study.

2) to include a short paragraph acknowledging that the AMOC evolution seen in the SENS experiment is intimately linked to the experimental design and will vary with relaxed assumptions such as repeat year, initial year, domain etc.

We have expanded the description of the SENS-configuration in the Methods section by providing a brief discussion (lines: 359 – 367) of this point which acknowledges the limitations of the experimental design.

A few minor comments:

Line 82: “Year to year variations in buoyancy forcing inhibited” – I am not totally convinced this is actually what is implemented (CORE II) even though this was the purpose. Some terms (e.g. evaporation) are likely still calculated internally from bulk formulations resulting in variations in year-to-year buoyancy forcing.

The formulation was indeed somewhat too lax: in principle, there are still some year-to-year variations in the buoyancy forcing, due to both the feedback terms in the bulk formulation and also the freshwater part of the buoyancy forcing. These effects are very small, however, compared to the dominating influence of the atmospheric

conditions (primarily, wind and temperature) on the buoyancy flux (as reflected, e.g., in the evolution of the mixed layer depth, Fig. 2a). In order to avoid distraction by too much technicalities, we propose to substitute the “inhibited” by “constrained”.

Line 288: reference missing I suspect.

The statement about the “lack of changes in ... the overflow” refers to ref. 39 given earlier (line 258). For clarity we have included it again at the end of the sentence here.

Reviewer #3

I'm happy with the author's revised version of their manuscript, both in terms of my earlier comments as well as the other reviewers. I have no further comments.